# Structure of human TRPV4 in complex with GTPase RhoA

Kirill D. Nadezhdin [1,5], Irina A. Talyzina [1,2,5], Aravind Parthasarathy [3], Arthur Neuberger [1], David X. Zhang [3,4] & Alexander I. Sobolevsky [1]

Transient receptor potential (TRP) channel TRPV4 is a polymodal cellular sensor that responds to moderate heat, cell swelling, shear stress, and small-molecule ligands. It is involved in thermogenesis, regulation of vascular tone, bone homeostasis, renal and pulmonary functions. TRPV4 is implicated in neuromuscular and skeletal disorders, pulmonary edema, and cancers, and represents an important drug target. The cytoskeletal remodeling GTPase RhoA has been shown to suppress TRPV4 activity. Here, we present a structure of the human TRPV4-RhoA complex that shows RhoA interaction with the membrane-facing surface of the TRPV4 ankyrin repeat domains. The contact interface reveals residues that are mutated in neuropathies, providing an insight into the disease pathogenesis. We also identify the binding sites of the TRPV4 agonist 4α-PDD and the inhibitor HC-067047 at the base of the S1-S4 bundle, and show that agonist binding leads to pore opening, while channel inhibition involves a π-to-α transition in the pore-forming helix S6. Our structures elucidate the interaction interface between hTRPV4 and RhoA, as well as residues at this interface that are involved in TRPV4 disease-causing mutations. They shed light on TRPV4 activation and inhibition and provide a template for the design of future therapeutics for treatment of TRPV4-related diseases.

Transient receptor potential (TRP) channel TRPV4 is a non-selective cation channel that is ubiquitously expressed in most tissues, including bone, skin, vasculature, heart, brain, lungs, liver, kidneys, and bladder[1-3]. The channel is activated by warm temperatures (>27 °C)[4], an increase of cell volume, endogenous ligands such as derivatives of arachidonic acid, and synthetic compounds[5-7]. Numerous previously characterized disease-causing mutations in different regions of TRPV4 were grouped in two major categories, causing skeletal dysplasia and motor/sensory neuropathies[6]. Most of the TRPV4 channelopathies are associated with gain-of-function mutations[7], providing strong motivation to study this channel's inhibition and gating. Several inhibitors of TRPV4 that display different affinity and selectivity were reported[8], the first being RN-1734

with 2-6 μM potency[9]. Using mutagenesis, binding of one inhibitor, HC-067047[10], was mapped to the region contributed by the S2-S3 linker, S4, and S5 helices[11], but the location of this site remains to be confirmed structurally. Among a few available TRPV4 agonists[8], the most known activators are the synthetic compounds 4α-phorbol 12,13-didecanoate (4α-PDD), GSK1016790A, and RN-1747[8], as well as the natural compounds anandamide and arachidonic acid[12].

TRPV4 is known to be regulated by intracellular proteins calmodulin, OS-9, STIM1, and PACSIN3[7,13,14]. In addition, a recent unbiased proteomics screening identified a small GTPase RhoA as an inhibitor of TRPV4, which in turn can be activated by TRPV4 through the ion channel activity and calcium influx[13,15]. RhoA is a member of the Ras

[1]Department of Biochemistry and Molecular Biophysics, Columbia University, New York, NY 10032, USA. [2]Integrated Program in Cellular, Molecular and Biomedical Studies, Columbia University, New York, NY 10032, USA. [3]Department of Medicine, Cardiovascular Center, Medical College of Wisconsin, Milwaukee, WI 53226, USA. [4]Department of Pharmacology and Toxicology, Medical College of Wisconsin, Milwaukee, WI 53226, USA. [5]These authors contributed equally: Kirill D. Nadezhdin, Irina A. Talyzina. ✉e-mail: as4005@cumc.columbia.edu

superfamily of GTPases[16], which is involved in processes of cell division, differentiation, proliferation, apoptosis and survival[17]. These enzymes act as molecular switches and can be in active GTP-bound form or inactive GDP-bound form[18]. Lipidation and targeting to membrane compartments were shown to be essential for the function of a majority of GTPases[19–21]. A subfamily of Rho GTPases, which includes RhoA, regulates cytoskeletal dynamics[22]. While functional interplay between TRPV4 and RhoA has been suggested to result from direct interaction between these proteins[13], the structure of their complex has been missing. Here we report the structure of the human TRPV4-RhoA complex, with RhoA binding to the membrane-facing side of the TRPV4 ankyrin repeat domain (ARD). We also solve structures of this complex bound to the agonist 4α-PDD and antagonist HC-067047 that shed light on the mechanisms of TRPV4 gating and inhibition.

## Results

### Functional and structural characterization of human TRPV4

To study the function of human TRPV4, we expressed it in HEK 293 cells and measured calcium uptake in response to the application of agonist 4α-PDD (Supplementary Fig. 1). Increasing concentrations of the agonist induced stronger changes in fluorescence yielding the concentration dependence of hTRPV4 activation by 4α-PDD with the half maximal effective concentration, $EC_{50} = 450 \pm 39$ nM ($n = 4$). We also studied hTRPV4 inhibition by applying 1 μM 4α-PDD in the presence of different concentrations of antagonist HC-067047. HC-067047 induced concentration-dependent inhibition of hTRPV4-mediated calcium uptake with the half-maximal inhibitory concentration, $IC_{50} = 25.0 \pm 6.2$ nM ($n = 4$).

For our structural studies, we expressed full-length wild-type human TRPV4 in HEK 293S cells and purified it in glyco-diosgenin

(GDN) detergent. First, we determined the structure of hTRPV4 in the apo state by subjecting the purified protein to single-particle cryo-EM in the absence of added ligands. The corresponding 3D reconstruction resulted in a 3.0-Å resolution structure (Supplementary Figs. 2, 3; Supplementary Table 1). The hTRPV4apo structure represents a tetramer assembled of four identical subunits, with the overall architecture reminiscent of other TRPV channels, which includes two main compartments: a transmembrane domain (TMD) with a central ion channel pore and an intracellular skirt where four subunits comprise walls enclosing a wide cavity underneath the ion channel (Fig. 1).

Each TRPV4 subunit consists of a disordered N-terminal region (residues 1–147), followed by an ankyrin repeat domain (ARD) with six ankyrin repeats (ARs), and a linker domain that includes a β-hairpin (composed of β-strands β1 and β2) and a helix-turn-helix motif resembling a seventh ankyrin repeat, and the pre-S1 helix, which connects the linker domain to the TMD (Supplementary Fig. 4). The TMD (residues 446–748) resembles TMDs in voltage-gated and other TRP channels and includes six transmembrane helices (S1–S6) and a pore loop (P-loop) between S5 and S6. The first four helices form a bundle called S1–S4 domain or voltage sensor-like domain (VSLD). S5, P-loop, and S6 comprise the pore domain that connects to the S1–S4 domain of the neighboring subunit in a domain-swapped arrangement. The relative positioning of S1–S4 and pore domains of hTRPV4 is similar to TRPV1-3,5-6 but somewhat different from the one observed in the structure of *Xenopus tropicalis* TRPV4 (Supplementary Fig. 5)[23]. The TMD ends with the amphiphilic TRP helix, which runs almost parallel to the membrane surface and represents a signature of the TRP channel family. Following the TRP helix, the polypeptide forms a short post-TRP helix and a loop structure named C-terminal hook, which continues with a third β-strand (β3) that tethers to the β-hairpin in the

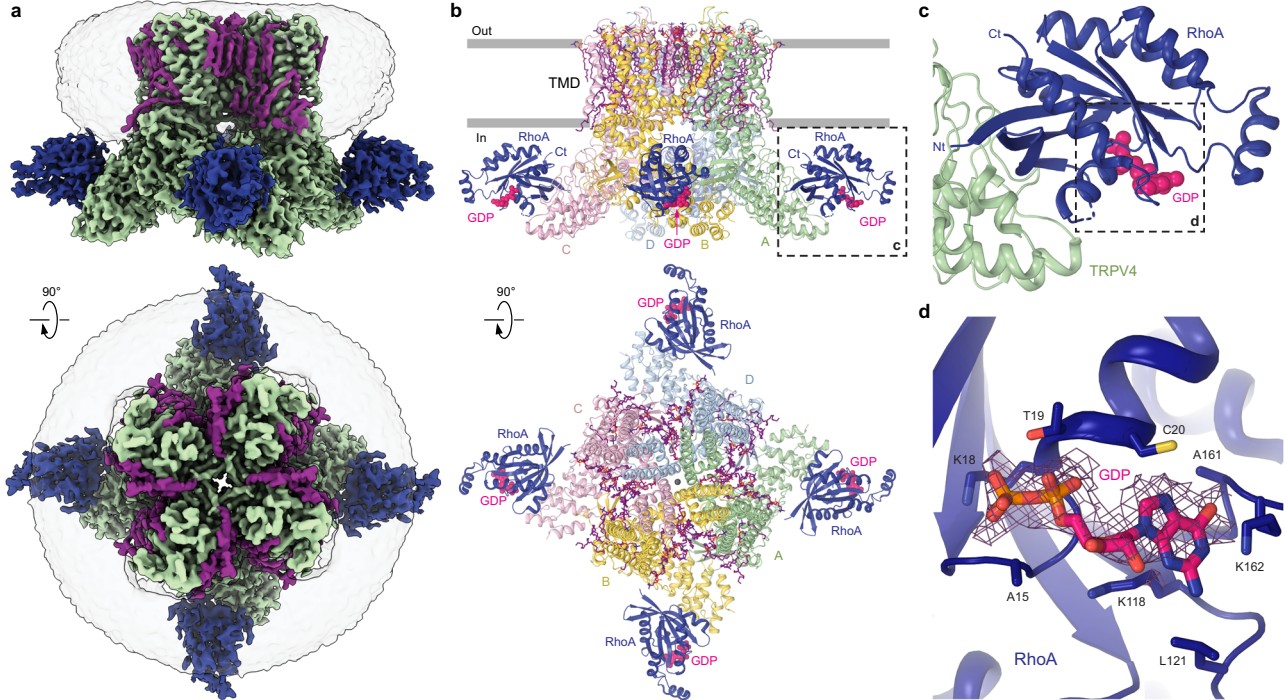

**Fig. 1 | Structure of human TRPV4 in complex with GTPase RhoA. a** Cryo-EM density map of the hTRPV4-RhoA complex in the apo state at 3.0 Å resolution, viewed parallel to the membrane (top) and extracellularly (bottom). hTRPV4 is colored green, lipid densities are purple, and RhoA is dark blue. The semi-transparent surface represents the micelle density. **b** TRPV4 structure with the subunits colored in green (subunit A), yellow (B), light pink (C), and blue (D). Lipids are shown as purple sticks. RhoA structure is colored dark blue. Guanosine diphosphate (GDP) molecules bound to RhoA are shown as space-filling models (bright pink). Grey spheres represent sodium ions. Dashed rectangle indicates the region expanded in panel (**c**). **c** Close-up view of the TRPV4-RhoA interaction interface. Dashed rectangle indicates the region expanded in panel (**d**). **d** Close-up view of the GDP binding pocket, with GDP (pink) and residues contributing to its binding shown in sticks. Purple mesh represents experimental density for GDP.

linker domain to create a three-stranded β-sheet. Following β3 is an extended region of the C-terminus that wraps around the three-stranded β-sheet and together with the C-terminal hook participates in intersubunit interactions with the ARDs that glue the elements of the intracellular skirt together.

Upon inspection, the cryo-EM map of hTRPV4$_{apo}$ revealed numerous non-protein densities around the TMD that represented annular lipids (Fig. 1, Supplementary Fig. 6). Four larger densities, one per TRPV4 subunit, were found adjacent to the ARs of the intracellular skirt. These densities were weaker than the density for the TRPV4 protein suggesting an underlying partial occupancy of the corresponding spaces. To improve the quality of this density, we performed symmetry expansion of the particles, followed by particle subtraction, local refinement, and 3D classification (Supplementary Fig. 2) and revealed two distinct groups of the hTRPV4 subunits, with and without a binding partner. The final high-resolution map of the hTRPV4 subunit with the binding partner allowed us to unambiguously build a model of the small GTPase RhoA, which is known to interact with TRPV4 channels in vivo[13]. It is important to note that we did not add RhoA cDNA or protein at any step of TRPV4 expression or purification. Thus, RhoA identified in our structure is an endogenous protein expressed in HEK 293S cells that was carried bound to hTRPV4 throughout purification. According to our data processing, RhoA binds to hTRPV4 at various stoichiometries, ranging from none to four RhoA molecules bound to a single hTRPV4 tetramer. Comparison of hTRPV4 subunits with and without bound RhoA suggests that the architecture of the individual hTRPV4 subunit does not undergo substantial changes in the presence of bound RhoA (Supplementary Fig. 7).

## Architecture of hTRPV4-RhoA complex

RhoA molecules bound to the ARDs of hTRPV4 are positioned adjacent to the membrane with their C-termini facing the membrane's intracellular leaflet (Fig. 1b). The fold of RhoA bound to hTRPV4 is nearly identical to the one observed in the previously published crystal structures of isolated RhoA[24]. The structure of RhoA consists of a six-stranded β-sheet surrounded by seven short helices connected by loops (Fig. 1c, Supplementary Fig. 4), and represents the fold conserved among Ras proteins and related small GTPases[25]. The quality of our cryo-EM map allowed us to unambiguously identify the endogenous molecule of guanosine diphosphate (GDP) bound in the RhoA ligand-binding pocket (Fig. 1c, d, Supplementary Fig. 6). The GDP binding pocket is the same as the one identified in GDP-bound crystal structures of RhoA[24]. In contrast to the crystal structures, we do not clearly see Mg$^{2+}$ in close proximity to GDP, possibly due to limited resolution of our cryo-EM reconstructions.

The interface between hTRPV4 and RhoA is formed by five loops connecting six ankyrin repeats of TRPV4 and β1, β2, β3 and α2 regions of RhoA (Fig. 1c, Supplementary Fig. 4). The strongest contacts are established between polar residues of TRPV4, including R232, R237, D263, R269, R315, and R316, and side chains and backbone carbonyls of RhoA residues R5, E40, E54, and D76 (Fig. 2). Remarkably, most residues involved in polar TRPV4-RhoA interactions are subjects of disease-causing mutations in TRPV4-linked channelopathies, including peripheral neuropathy and skeletal dysplasia[15,26,27] (Fig. 2). Consistent with our structure, previous studies showed that neuropathy-related mutants R232C, R237L, R269C, and R315W disrupt protein-protein interactions between TRPV4 and RhoA[13]. Mutations of the residues R5 and E40 in RhoA, which form salt bridges with D263 and R232 in TRPV4, respectively (Fig. 2c), have been associated with different types of cancer[28]. In addition, we introduced the R316A mutation that resulted in increased baseline calcium uptake, consistent with previous studies of neuropathy-causing mutations at the TRPV4-RhoA interface[13], and dramatically weakened activation by 4α-PDD (Supplementary Fig. 1f–h), providing verification of our structure and strongly supporting the inhibitory role of RhoA in hTRPV4 function.

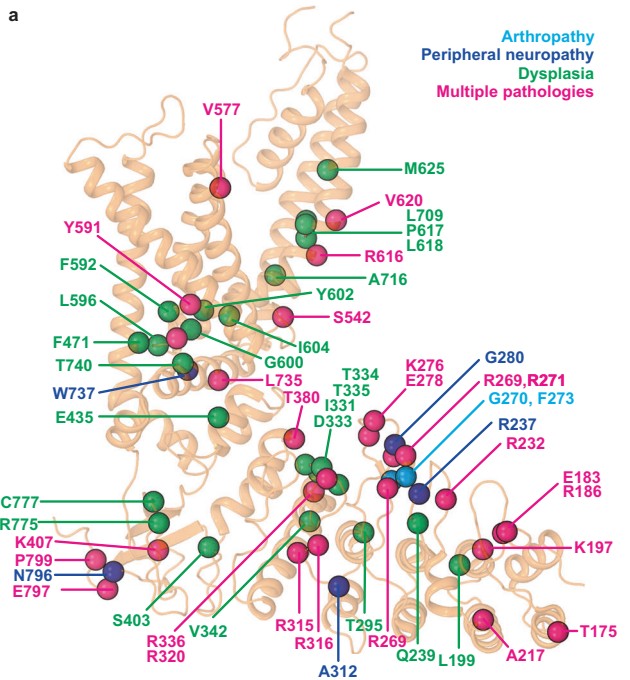
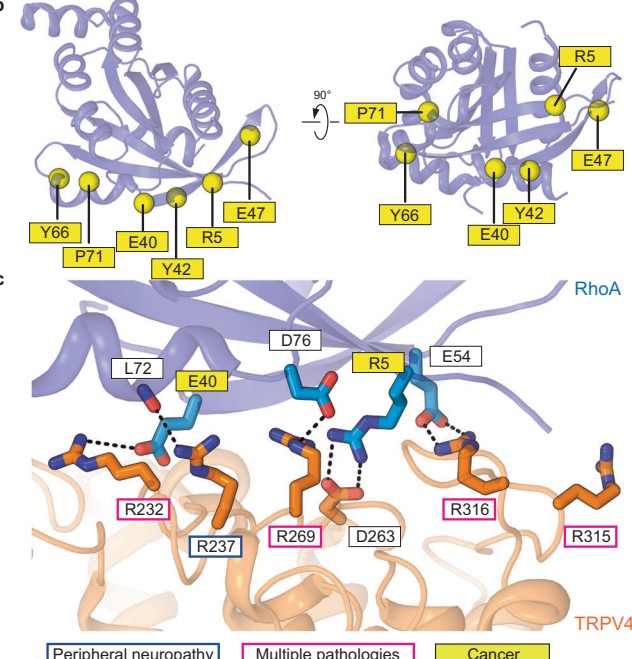

**Fig. 2 | TRPV4 channelopathies and cancer mutations in RhoA. a** TRPV4-linked disease-associated mutations mapped onto the structure of a single hTRPV4$_{apo}$ subunit (orange ribbon). Residue positions for disease-causing mutations in dysplasia (skeletal, metatropic, Spondylometaphyseal, or Kozlowski type), peripheral neuropathy (Charcot-Marie-Tooth disease type 2) and arthropathy (Familial digital arthropathy-brachydactyly) are shown as green, dark blue, and light blue spheres, respectively. Mutations causing multiple pathologies (neuropathy, dysplasia, arthropathy, etc.) are colored pink. **b** Mutations in RhoA associated with cancer. **c** TRPV4-RhoA interface with the interacting residues shown in sticks. Residues undergoing disease-causing mutations are labeled. Dashed lines represent putative ionic interactions.

## Closed pore of hTRPV4 in the apo state

A closer look at the ion channel pore in hTRPV4$_{apo}$ revealed two constriction points, in the upper pore or the selectivity filter and at the lower pore or the gate region, typical for the tetrameric ion channels (Fig. 3a). The selectivity filter constriction is formed by the backbone carbonyl oxygens of glycines G679, with the cross-pore distance of 6.6 Å between the diagonal pairs, permissive for conductance of hydrated sodium or potassium ions. The narrowest point of the gate region is formed by isoleucines I715. With the 10.2 Å distance between the Cα atoms of the diagonally positioned I715, the side chains of these isoleucines face the center of the pore, likely forming a tight hydrophobic seal and making the pore impermeable to water or ions. Confirming the closed conformation of the ion channel pore in hTRPV4$_{apo}$, measurements of the pore radius show that at the level of I715 it is smaller than the radius of the water molecule (Fig. 3b). Interestingly, in this non-conducting apo state of hTRPV4, the pore-forming S6 helices contain a π-bulge, signifying the region undergoing an α-to-π transition in S6 during gating in other TRP channels[29–32].

## TRPV4 structure in complex with agonist 4α-PDD

To get insight into the gating mechanism of hTRPV4, we subjected the purified protein to single-particle cryo-EM in the presence of the 4α-PDD agonist (Fig. 4; Supplementary Fig. 8). Cryo-EM data processing revealed two populations of particles, both bound to RhoA. One population resulted in a 3D reconstruction identical to hTRPV4$_{apo}$, while the second yielded a conformationally different 3.35-Å resolution structure hTRPV4$_{4αPDD}$, with bulky densities revealing four 4α-PDD binding sites, one per subunit, at the base of the S1-S4 bundle (Fig. 4a, b, Supplementary Fig. 6d). The S1-S4 bundle is an important site for the allosteric modulation of vanilloid subfamily TRP channels which we previously reported as the allosteric binding site for TRPV3 activation by the small synthetic compound 2-aminoethoxydiphenyl borate (2-APB)[30] as well as its inhibition by the plant-derived coumarin osthole[33]. The hTRPV4$_{4αPDD}$ structure, which was refined with C2 symmetry, displayed a slight deviation from the 4-fold symmetry observed for hTRPV4$_{apo}$. Nevertheless, all four 4α-PDD binding sites in

the hTRPV4$_{4αPDD}$ structure were very similar and included residues S470, N474, S477, F524, N528, Y553, Y591, D743, I744, S747 and F748 (Fig. 4b). Previously, the N474Q mutant showed more than 100-fold increase in the $EC_{50}$ value for 4α-PDD-induced activation compared to the wild-type channel[34]. In addition, we introduced the N474A, D546A and Y591A mutations in the 4α-PDD binding pocket and found various degree alterations in the 4α-PDD potency, likely reflecting the extent of the corresponding residue involvement into 4α-PDD binding (Supplementary Fig. 1f). Overall, ours as well as previous studies combining mutagenesis and functional analysis strongly confirm the 4α-PDD binding sites identified in the hTRPV4$_{4αPDD}$ structure.

In 4α-PDD-bound structure, TRPV4 maintained a similar overall shape and RhoA binding was nearly identical to those in hTRPV4$_{apo}$. However, binding of 4α-PDD resulted in dramatic changes in the pore of hTRPV4$_{4αPDD}$ (Fig. 4c, Supplementary Movie 1). The narrow regions of both the selectivity filter and the gate were greatly expanded compared to the corresponding regions in the apo-state structure (Fig. 3a). The dramatic widening of the pore was confirmed by measurements of its radius using HOLE (Fig. 3b) and suggested that hTRPV4$_{4αPDD}$ may represent an open conducting state of hTRPV4. The opening of the channel in hTRPV4$_{4αPDD}$ was also accompanied by movement of the intracellular skirt towards the membrane and transformation of the C-terminus that wraps around the 3-stranded β-sheet in hTRPV4$_{apo}$ into an α-helix (Supplementary Fig. 9), the characteristic changes previously seen during TRPV3 gating[35–37].

For the hTRPV4$_{4αPDD}$ cryo-EM sample preparation, we supplemented the protein with the slowly hydrolysable GTP analog GTPγS. Nevertheless, the nucleotide density was resolved poorly, similar to the presumed GDP density in hTRPV4$_{apo}$ (Fig. 1d), making it unclear whether GTPγS did bind RhoA. Additional experiments are required to study the effects of nucleotides on RhoA regulation of hTRPV4.

## TRPV4 structure in complex with antagonist HC-067047

To study the molecular mechanism of TRPV4 inhibition, we solved the structure of TRPV4 in complex with the highly potent small-molecule inhibitor HC-067047. We purified the TRPV4 protein and

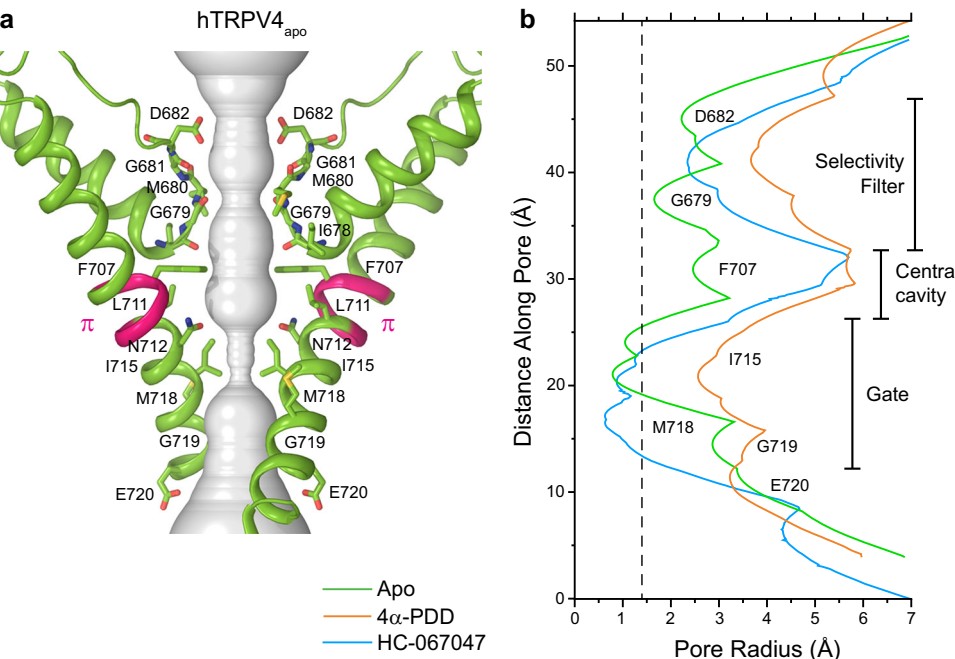

**Fig. 3 | Pore geometry in hTRPV4$_{apo}$ and other structures. a** Pore-forming domain in hTRPV4$_{apo}$ with the residues contributing to pore lining shown as sticks. Only two of four subunits are shown, with the front and back subunits omitted for clarity. The pore profile is shown as a space-filling model (grey). The region that

undergoes the α-to-π transition in S6 is labeled (π). **b** Pore radius for hTRPV4$_{apo}$ (green), hTRPV4$_{4αPDD}$ (orange), and hTRPV4$_{HC}$ (blue) calculated using HOLE. The vertical dashed line denotes the radius of a water molecule, 1.4 Å.

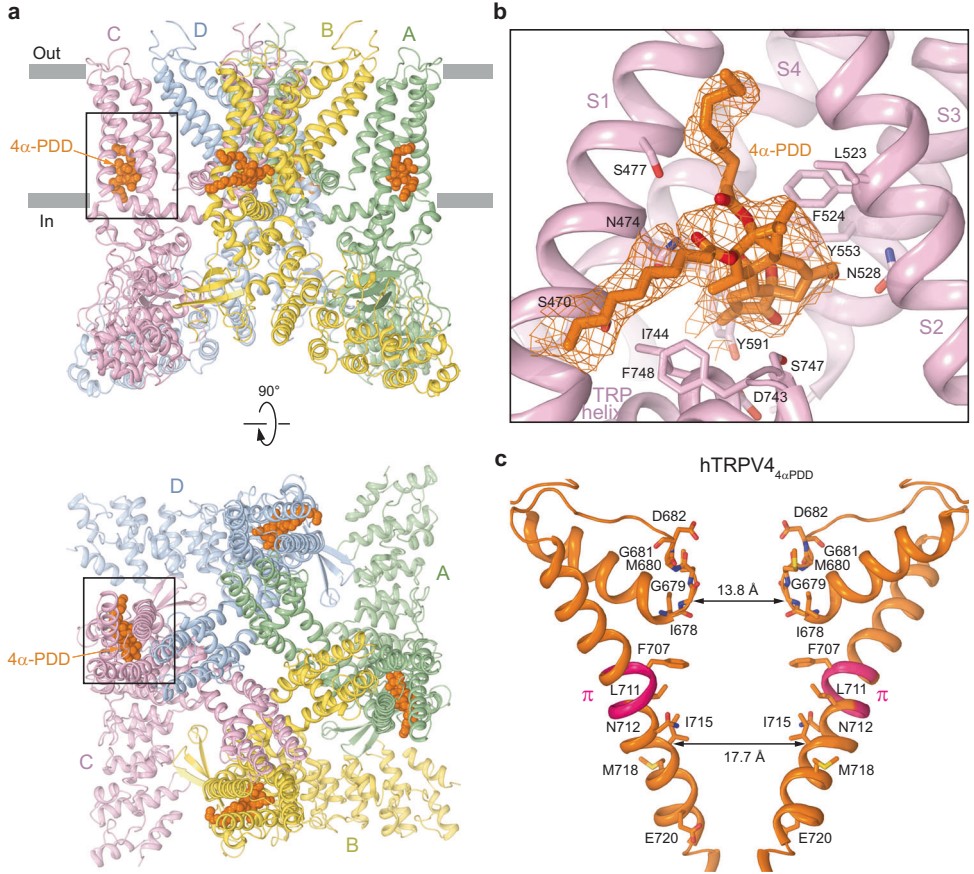

**Fig. 4 | Structure of hTRPV4 in complex with the agonist 4α-PDD.**
**a** hTRPV4$_{4αPDD}$ structure viewed parallel to the membrane (top) or extracellularly (bottom), with subunits colored green, yellow, pink, and blue, and molecules of 4α-PDD shown as space-filling models (orange). **b** Close-up view of the binding pocket, with 4α-PDD (orange) and residues that contribute to its binding shown in sticks. Orange mesh represents experimental density for 4α-PDD. **c** Pore-forming domain

in hTRPV4$_{4αPDD}$ with the residues contributing to pore lining shown as sticks. Only two (A and C) of four subunits are shown, with the front and back subunits omitted for clarity. The distances between Cα atoms of residues at the pore constrictions in the selectivity filter (G679) and gate (I715) regions are indicated. The region that undergoes the α-to-π transition in S6 (pink) is labeled (π).

subjected it to single-particle cryo-EM in the presence of 200 μM HC-067047. Cryo-EM data analysis revealed a population of particles that yielded a HC-067047-bound structure, TRPV4$_{HC}$ (Fig. 5; Supplementary Fig. 10). Similar to hTRPV4$_{4αPDD}$, we did not observe differences in RhoA conformation or its binding to the channel in TRPV4$_{HC}$ compared to hTRPV4$_{apo}$.

The overall architecture of hTRPV4$_{HC}$ resembled the one of the apo and agonist-bound structures (Fig. 5a). Like the agonist-bound hTRPV4$_{4αPDD}$ structure, hTRPV4$_{HC}$ showed a slight deviation from the 4-fold symmetry. Inspection of the cryo-EM map revealed four densities shaped as HC-067047, one per subunit, located at the base of the S1-S4 bundle, the site that mediates binding and allosteric inhibition of TRPV3 by the plant-derived coumarin osthole[33]. To improve the quality of the cryo-EM reconstruction at this site, we performed a local refinement of the region covering the S1-S4 bundle, TRP helix, and linker domain. As a result, the improved quality of the cryo-EM map in this region allowed unambiguous fit of the HC-067047 molecule into the corresponding density (Supplementary Fig. 6d) and identification of side chains for the residues contributing to its binding (Fig. 5b). Similar to 4α-PDD, HC-067047 resides right above the TRP helix, being bound to the S1-S4 bottom site[38] and surrounded by the side chains of F471, N474, S477, Y478, D546, Y553, Y591, F592, and D743. Strongly confirming the HC-067047 binding site, the N474A, D546A and Y591A mutations caused substantial reduction in the HC-067047 potency of hTRPV4 inhibition (Supplementary Fig. 1g).

The conformational changes that accompany HC-067047 binding are especially noticeable in the pore region. They included a π-to-α transition in the middle of the S6 helix, which eliminated the π-bulge (residues F707-L711), caused a ~100° rotation of the C-terminal part of S6 and made it entirely α-helical (Fig. 5c, Supplementary Movie 2). As a result, the side chains of four M718 residues created a hydrophobic seal of the pore. In addition, the hTRPV4$_{HC}$ selectivity filter was substantially dilated compared to TRPV4$_{apo}$, emphasizing strong structural changes induced by binding of the inhibitor. Interestingly, HC-067047 binding did not cause the movement of the intracellular skirt towards the membrane or the C-terminus unwrapping that were observed in response to binding of the agonist 4α-PDD, making these regions look similar to the apo state structure hTRPV4$_{apo}$ (Supplementary Fig. 9).

## Discussion
TRPV4 channel is a polymodal sensor that resides in the plasma membrane, being activated by a variety of physical and chemical stimuli including changes in temperature and membrane stretch. While it was shown that mice deficient in TRPV3 and TRPV4 exhibited no obvious alterations in the thermal preference[39], knockout of TRPV4 caused the late-onset of hearing loss[40] and impaired pressure sensation[41]. Many inherited alterations of the *TRPV4* gene have been identified as gain-of-function mutations, often leading to disorders of the peripheral nervous or skeletal systems[26]. Accordingly, pharmacological modulation of the channel function is a potential strategy for treating TRPV4-linked diseases.

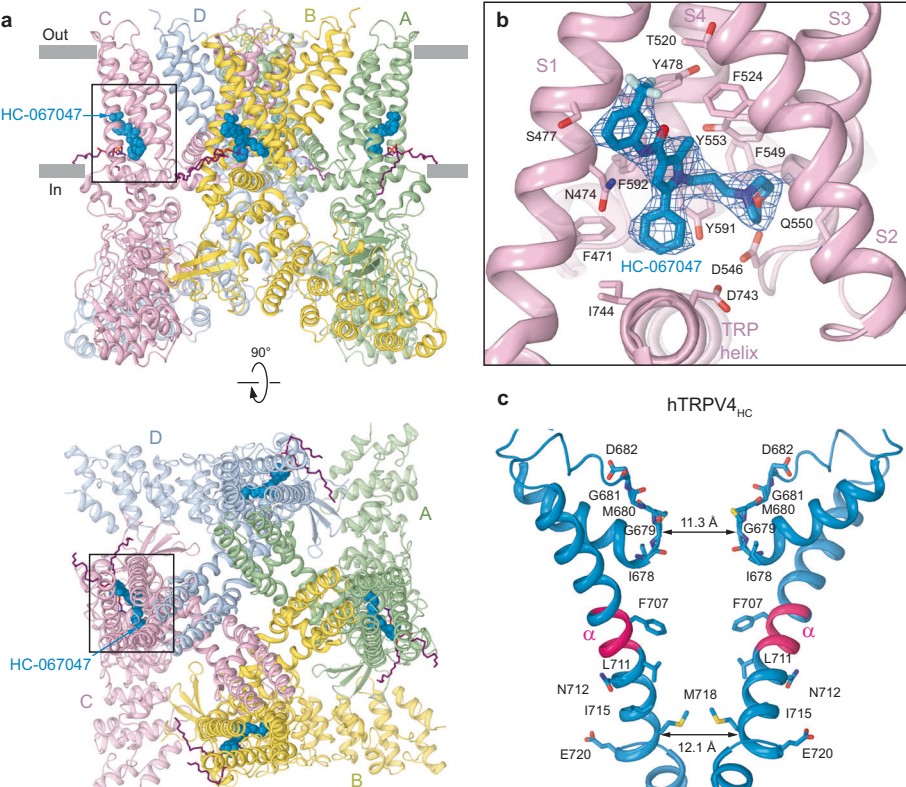

**Fig. 5 | Structure of hTRPV4 in complex with the inhibitor HC-067047.**
**a** hTRPV4$_{HC}$ structure viewed parallel to the membrane (top) or extracellularly (bottom), with subunits colored green, yellow, pink, and blue, and lipids shown as purple sticks. Molecules of HC-067047 are shown as space-filling models (blue). **b** Close-up view of the binding pocket, with HC-067047 (blue) and residues that contribute to its binding shown in sticks. Blue mesh represents experimental density for HC-067047. **c** Pore-forming domain in hTRPV4$_{HC}$ with the residues contributing to pore lining shown as sticks. Only two (A and C) of four subunits are shown, with the front and back subunits omitted for clarity. The distances between Cα atoms of residues at the pore constrictions in the selectivity filter (M680) and gate (M718) regions are indicated. The region that undergoes the π-to-α transition in S6 (pink) is labeled (α).

The TRPV4 function is regulated through ARDs by many endogenous small molecules and proteins, including calmodulin and RhoA[13,42,43]. It was shown that ATP and calmodulin binding did not change the ARDs overall structure[44–47], confirming that structural stability is an intrinsic property of ARDs. Although the impact of ARDs on TRPV channel behavior is not fully understood, it was demonstrated previously that the function of TRPV3 channel, closely related to TRPV4, is influenced by the movement of the entire ARD domain relative to the rest of the channel[35]. While calmodulin potentiates TRPV4 channel opening, the TRPV4-RhoA interaction has been proposed to mediate the reciprocally inhibiting actions of the two proteins, where binding of RhoA inhibits TRPV4 and binding of TRPV4 inhibits RhoA. While it has been shown that RhoA in the inhibited state binds TRPV4, dissociation of RhoA from TRPV4 appears to be insufficient to activate RhoA[13]. On the other hand, increased cell volume elevates TRPV4-mediated $Ca^{2+}$ influx[14]. Interestingly, an increase in the intracellular concentration of $Ca^{2+}$ has been found to promote a sustained activation of RhoA[48], potentially linking TRPV4 opening to the change in the state of RhoA activation. In addition, RhoA activation was shown to be necessary for the actin fiber-mediated mechanotransduction via the LIM domain-containing proteins[49]. Specifically, RhoA activates the regulatory protein testin, which in turn enables LIM domain to bind stretched actin filaments[50]. Overall, the TRPV4-RhoA complex can serve as a link between the rapid and slow responses to mechanical stretch: TRPV4 opening mediates fast calcium influx that causes sustained RhoA activation, which in turn potentiates mechanosensation via stretched actin filaments. Further studies, including acquisition of structural data under the membrane stretched conditions, are necessary to better understand the mechanisms of mechanosensation[51–54] and the role of TRPV4 and GTPase RhoA interaction in regulation of cellular functions.

While our manuscript was in preparation, another manuscript describing the TRPV4-RhoA interaction has been published on BioRxiv (https://doi.org/10.1101/2023.03.15.532784). The results of this manuscript are largely consistent with our findings.

## Methods

### Constructs

For structural studies, cDNA for full-length human *TRPV4* (NM_021625) was introduced into pEG-BacMam vector for protein expression in mammalian cells[55], with C-terminal region coding for the thrombin cleavage site (residues LVPRG), followed by the green fluorescent protein (GFP) and an the streptavidin affinity tag (residues WSHPQFEK), as described before[56,57].

### Protein expression and purification

hTRPV4 bacmids and baculoviruses were produced using standard procedures[56]. Briefly, baculovirus was made in Sf9 cells (Thermo Fisher Scientific, mycoplasma test negative, GIBCO #12659017) for ~96 h and added to suspension-adapted HEK 293S cells lacking N-acetyl-glucosaminyltransferase I (GnTI⁻, mycoplasma test negative, ATCC #CRL-3022) that were maintained at 37 °C and 6% $CO_2$ in Freestyle 293 media (Gibco-Life Technologies #12338-018) supplemented with 2% FBS. To reduce hTRPV4 cytotoxicity, 10 μM ruthenium red was added to the suspension of HEK 293S cells. To enhance protein expression, sodium butyrate (10 mM) was added 24 h after transduction, and the temperature was reduced to 30 °C. The cells were harvested 48 h after transduction by 15-min centrifugation at 5471 × *g* using a Sorvall

Evolution RC centrifuge (Thermo Fisher Scientific). The cells were washed in the phosphate-buffered saline (PBS) pH 8.0 and pelleted by centrifugation at 3202 × $g$ for 10 min using an Eppendorf 5810 centrifuge.

hTRPV4 was purified based on our previously established protocols[33,56,58–60]. In a nutshell, the cell pellet was resuspended in the ice-cold buffer containing 20 mM Tris pH 8.0, 150 mM NaCl, 0.8 μM aprotinin, 4.3 μM leupeptin, 2 μM pepstatin A, 1 μM phenylmethylsulfonyl fluoride (PMSF), and 1 mM β-mercaptoethanol (βME). The suspension was supplemented with 1% (w/v) glycodiosgenin (GDN), and cells were lysed at constant stirring for 1 h at 4 °C. Unbroken cells and cell debris were pelleted in the Eppendorf 5810 centrifuge at 3202 × $g$ and 4 °C for 10 min. Insoluble material was removed by ultracentrifugation for 1 h at 186,000 × $g$ in a Beckman Coulter centrifuge using a 45 Ti rotor. The supernatant was added to the strep resin, which was then rotated for 20 min at 4 °C. The resin was washed with 10 column volumes of wash buffer containing 20 mM Tris pH 8.0, 150 mM NaCl, 1 mM βME, and 0.01% (w/v) GDN, and the protein was eluted with the same buffer supplemented with 2.5 mM D-desthiobiotin. The eluted protein was concentrated to 0.5 ml using a 100-kDa NMWL centrifugal filter (MilliporeSigma™ Amicon™) and then centrifuged in a Sorvall MTX 150 Micro-Ultracentrifuge (Thermo Fisher Scientific) for 30 min at 66,000 × $g$ and 4 °C using a S100AT4 rotor before injecting it into a size-exclusion chromatography (SEC) column. The protein was purified using a Superose™ 6 10/300 GL SEC column attached to an AKTA FPLC (GE Healthcare) and equilibrated with the buffer containing 150 mM NaCl, 20 mM Tris pH 8.0, 1 mM βME, and 0.01% (w/v) GDN. The tetrameric peak fractions were pooled and concentrated to 1.6–3.2 mg/ml using a 100-kDa NMWL centrifugal filter (MilliporeSigma™ Amicon™).

## Cryo-EM sample preparation and data collection

Prior to sample application, UltrAuFoil R 1.2/1.3 (Au 300) grids were plasma treated in a PELCO easiGlow glow discharge cleaning system (0.39 mBar, 15 mA, "glow" for 20 s, and "hold" for 10 s). A Mark IV Vitrobot (Thermo Fisher Scientific) set to 100% humidity at 22 °C (hTRPV4 in the apo state or in the presence of 4α-PDD) or 4 °C (hTRPV4 in the presence of HC-067047) was used to plunge-freeze the grids in liquid ethane after applying 3 μl of protein sample to their gold-coated side using the blot time of 3 s, blot force of 3, and wait time of 15 s. The grids were stored in liquid nitrogen before imaging. To determine the inhibitor-bound structure, the protein was supplemented with 200 μM of HC-067047 (dissolved in DMSO) and incubated at room temperature for 10 min before grid freezing. To solve the agonist-bound structure, the protein was supplemented with 200 μM of 4α-PDD (dissolved in DMSO) and 10 mM GTPγS right before grid freezing.

Images of frozen-hydrated particles of hTRPV4 in the apo state were collected using the Leginon[61] software on a Titan Krios transmission electron microscope (TEM) (Thermo Fisher Scientific) operating at 300 kV and equipped with a post-column GIF Quantum energy filter and a Gatan K3 direct electron detecting (DED) camera (Gatan, Pleasanton, CA, USA). A total of 5456 micrographs were collected in the counting mode with an image pixel size of 0.83 Å across the defocus range of −0.5 to −1.5 μm. The total dose of ~58 e⁻Å⁻² was attained by using the dose rate of ~16 e⁻ pixel⁻¹s⁻¹ across 50 frames during the 2.5-s exposure time.

Images of frozen-hydrated particles of hTRPV4 in the presence of 200 μM HC-067047 were collected using Leginon software on a Titan Krios TEM operating at 300 kV and equipped with a post-column GIF Quantum energy filter and a Gatan K3 direct DED camera. A total of 11,551 micrographs were collected in the counting mode with an image pixel size of 0.83 Å across the defocus range of −0.5 to −1.5 μm. The total dose of ~58 e⁻Å⁻² was attained by using the dose rate of ~16 e⁻ pixel⁻¹s⁻¹ across 50 frames during the 2.5-s exposure time.

Images of frozen-hydrated particles of hTRPV4 in the presence of 200 μM 4α-PDD and 10 mM GTPγS were collected using SerialEM software on a Titan Krios TEM operating at 300 kV and a Gatan K3 direct DED camera. A total of 15,624 micrographs were collected in the counting mode with an image pixel size of 0.7888 Å across the defocus range of −0.75 to −1.5 μm. The total dose of ~60 e⁻Å⁻² was attained by using the dose rate of ~15.5 e⁻ pixel⁻¹s⁻¹ across 50 frames during the 2.4-s exposure time.

## Image processing and 3D reconstruction

Cryo-EM data were processed in cryoSPARC 4.2.0[62] and Relion 4.0[63]. For example, for hTRPV4 in the apo state, movie frames were aligned using the MotionCor2 algorithm implemented in Relion 4.0. The contrast transfer function (CTF) estimation was performed using the patch CTF. Following CTF estimation, micrographs were manually inspected and those with outliers in defocus values, ice thickness, and astigmatism as well as micrographs with lower predicted CTF-correlated resolution (higher than 6 Å) were excluded from further processing (individually assessed for each parameter relative to the overall distribution). The total number of 1,285,028 particles were picked using internally generated 2D templates and extracted with the 320-pixel box size and then binned to the 128-pixel box size. After several rounds of reference-free 2D classifications and Heterogeneous Refinements in cryoSPARC 4.2.0 with one reference class and three automatically generated "garbage" classes, the best 232,868 particles were imported in Relion and re-extracted with the 320-pixel box size without binning. These particles were subjected to one round of 3D classification without imposing symmetrical restraints (C1 symmetry) into 10 classes ($T = 4$). At this point, two clusters of well-defined classes were observed, one with C4-symmetrical pore domain and another one with a C2-symmetrical pore region. Particles representing the C4 cluster were combined and 3D classified (C1 symmetry, 6 classes, $T = 8$), and particles for the best class were subjected to CTF refinements to correct for beam tilt, higher order aberrations, anisotropic magnification, per particle defocus, and per micrograph astigmatism[64]. CTF-refined particles were subjected to Bayesian polishing and CTF refined again using the same procedure as described above. Polished and CTF-refined particles were imported into cryoSPARC 4.2.0. The final set of 37,266 particles representing the best C4 class was subjected to homogenous, non-uniform, and CTF refinements with C4 rotational symmetry. The reported resolution of 3.00 Å for the final map was estimated using the gold standard Fourier shell correlation (GSFSC). To improve the quality of reconstruction for the N-terminal ankyrin repeats region and RhoA, the final set of particles was C4 symmetry expanded, and the N-terminal domain region with RhoA was extracted from the 2D images of individual particles by subtraction of other regions. Subtracted particles comprising only the ARD-RhoA region were subjected to local refinement with a mask around this region and subjected to 3D classification without alignment in cryoSPARC 4.2.0. The final set of 86,562 subtracted particles representing the best classes with resolved secondary structure was subjected to the local refinement with the same mask and yielded a map with the resolution of 3.49 Å. The resolution of the maps was estimated using the FSC = 0.143 criterion[65]. Cryo-EM density were visualized using UCSF ChimeraX[66].

The dataset containing 4α-PDD was subjected to processing analogous to that described above for the apo state, with the exception that the entire processing was conducted using cryoSPARC 4.2.1[62]. First, 15,624 movie stacks were aligned using Patch Motion Correction and then subjected to Patch CTF estimation. The obtained micrographs were semi-manually curated based on their relative ice thickness and CTF fit resolution, with micrographs that had a predicted resolution higher than 6 Å being excluded. Particles were initially picked using templates generated from the TRPV4_apo map and then used to train the Topaz algorithm that picked a total of 3,935,964

particles. These particles were extracted using a 320-pixel box size and subsequently binned to a 96-pixel box size for cleaning through several rounds of 2D classification and Heterogeneous Refinement, similar to the processing of TRPV4$_{apo}$. A final subset of 373,164 particles was used for non-uniform refinement with imposed C2-symmetry, and this reconstruction was used for global and local CTF refinement, followed by additional non-uniform refinement. The particles were then divided into 8 classes using 3D classification without alignment, and the maps obtained from the three best-resolved and one worst-resolved classes were used for template-based 3D classification without alignment. This classification was conducted with a mask that covered only the transmembrane domain and a high-pass resolution set to 15 Å in order to diminish the micelle contribution. The result of this classification was a well-resolved apo state without densities for 4α-PDD (75,146 particles, 2.77 Å, refinement in C4) and two similar open states with C2 symmetry: a better-resolved state with more distantly located opposite subunits (80,823 particles, 3.35 Å, refinement in C2) and a state with slightly less separated opposite subunits and lower resolution (72,405 particles, 3.93 Å, refinement in C2). Due to the similarity of these two open states, only the map with better resolution was used for model building. In order to obtain a better density for the ligand bound inside the cavity formed by S1–S4 bundle and TRP helix in the open state, we conducted local refinement for this region. Because the open state map had C2 symmetry and two types of subunit conformation, local refinement was carried out separately for each pair of diagonal subunits. To further improve the density of the S1–S4 region, a focused classification was performed after local refinement with a spherical mask covering the binding site. Classes containing non-protein densities inside the binding pocket were subjected to local refinement with a mask covering the S1–S4 bundle and TRP helix, and the resulting maps with the most prominent densities were used for model building.

To reveal structural changes induced by RhoA binding, local refinement of one subunit was performed for the apo state class from the 4α-PDD containing dataset, due to its higher resolution. Prior to local refinement of the subunit, particles were subjected to symmetry expansion using the C4 symmetry. To obtain a subset of particles with full and minimal RhoA occupancy, 3D classification was performed with a mask covering only RhoA and target resolution set to 20 Å. Additionally, a subset of particles with full RhoA occupancy was used for local refinement with a mask covering ARD and RhoA in order to improve the quality of the RhoA density.

The dataset collected in the presence of HC-067047 was processed in a similar manner as the 4α-PDD dataset, using cryoSPARC 4.2.1[62]. Initially, 11,551 movie stacks were aligned using Patch Motion Correction followed by CTF estimation. Only micrographs with predicted resolution greater than 6 Å were used for template-based particle picking, with the TRPV4$_{apo}$ map used as a template. The particles were then extracted with a 320-pixel box size and binned to 128-pixel box size for cleaning, which was done by multiple rounds of 2D classification and heterogeneous refinement. The resulting subset of 340,855 particles was re-extracted with the full 320-pixel box size and used for non-uniform refinement with imposed C2 symmetry. After global and local CTF correction, we ran a second non-uniform refinement, yielding a 3.23 Å map. Subsequently, a 3D classification without alignment was run with a mask covering the transmembrane domain, yielding two classes with clear C2 symmetry but differing in the conformation of the transmembrane domain. One of these classes had a poorly resolved selectivity filter and was excluded from the final reconstructions, while the other class had a close to four-fold symmetrical shape with four densities corresponding to HC-067047 and was used for model building. This subset contained 65,881 particles and yielded a 3.49 Å 3D reconstruction. To improve the HC-067047 density, local refinement was performed on the S1–S4 bundle with the TRP helix. This involved using four subunits of the four-fold-like class and two opposite subunits of a class with a distorted pore. The resulting 3D reconstruction was done on 450,320 particles after particle subtraction and local refinement and yielded a 3.27-Å 3D reconstruction.

## Model building

The core of the hTRPV4 model was built in Coot[67] using the previously published cryo-EM structure of hTRPV4 (PDB ID: 7AA5)[34] as a guide. Regions that were not present in the previously published structure were built de novo, using the cryo-EM density as a guide. RhoA model was built using the previously published X-ray structure of RhoA (PDB ID: 1FTN)[24] as a guide. Other hTRPV4 structures were built using TRPV4$_{apo}$ as a guide. The models were tested for overfitting by shifting their coordinates by 0.5 Å (using Shake) in Phenix[68], refining each shaken model against the corresponding unfiltered half map, and generating densities from the resulting models in UCSF ChimeraX. The resulting models were real space refined in Phenix 1.18 and visualized using UCSF ChimeraX, and PyMOL (The PyMOL Molecular Graphics System, Version 2.0 Schrödinger, LLC.). The pore radius was calculated using HOLE[69].

## Cell preparation for calcium imaging experiments

HEK 293 cells between passages 6 to 10, were provided by Dr. David Wilcox (Medical College of Wisconsin) and grown in Dulbecco's Modified Eagle Medium (DMEM) supplemented with 10% FBS, 1 × PSG (100 units/ml penicillin G, 100 μg/ml streptomycin, and 2 mM glutamine), and incubated at 37 °C with 5% CO$_2$. For the present study, HEK 293 cells were used between passages 12 and 15.

The full-length human *TRPV4* (NM_021625) cDNA clone was obtained from OriGene (Rockville, MD) and shuttled into a mammalian expression vector resulting in a TRPV4 fusion protein with its COOH terminus tagged with turbo GFP. The TRPV4-GFP construct was then cloned into the pWPTS lentiviral vector as described previously[70]. All constructs were verified by DNA sequencing. For transient expression, HEK 293 cells were plated in 35-mm petri dishes and transfected with 0.5 μg plasmid DNA/35-mm dish using Lipofectamine 2000 reagent (Invitrogen) according to the manufacturer's protocol. To minimize potential Ca$^{2+}$ overload in the cells transfected with TRPV4 channels, a reduced-Ca$^{2+}$ media were used after transfection, with the Ca$^{2+}$ concentration decreased to ~0.6 mM by the addition of 1.2 mM EGTA, and the pH readjusted with NaOH. Cells were grown for an additional 16–20 h prior to Ca$^{2+}$ imaging.

## Fura-2 calcium imaging

HEK 293 cells were transfected with human TRPV4-GFP wild-type or mutant plasmids in 35-mm glass-bottom Petri dishes and grown to 60–70% confluence. Cells were incubated with fura-2 AM (5 μM) (Molecular Probes) and 0.02% Pluronic F-127 at 37 °C for 30 min in the modified Hank's balanced salt solution (HBSS) that contained (in mM): 123 NaCl, 5.4 KCl, 1.6 CaCl$_2$, 0.5 MgCl$_2$, 0.4 MgSO$_4$, 4.2 NaHCO$_3$, 0.3 Na$_2$HPO$_4$, 0.4 KH$_2$PO$_4$, 5.5 glucose, 20 HEPES, pH 7.4 with NaOH. Fura-2 assay was used to monitor cytosolic Ca$^{2+}$ signals as described previously[70]. The emitted fura-2 fluorescence at 510 nm in cells that are alternately exposed to 340 and 380 nm excitation wavelength was recorded and analyzed by the MetaFluor software (Version 7.10, Molecular Devices). Changes in the intracellular Ca$^{2+}$ concentration ([Ca$^{2+}$]$_i$) were presented as the ratio of the fluorescence intensity at 340 nm versus 380 nm excitation (F340/F380). For each experiment, 20-40 cells were selected with MetaFluor based on the basal [Ca$^{2+}$]$_i$. The cells with high basal [Ca$^{2+}$]$_i$ (F340/F380 ratio > 2.0) were excluded to minimize potential variations in response to TRPV4 agonists. GFP-negative cells from the same dish were used as controls. The images of cell fluorescence were acquired every 3 s, with background fluorescence subtracted before the experiment. The F340/380 ratios were averaged and plotted from at least 3 independent experiments.

## Reporting summary

Further information on research design is available in the Nature Portfolio Reporting Summary linked to this article.

## Data availability

All data are available from the corresponding authors upon request. The cryo-EM maps have been deposited in the Electron Microscopy Data Bank with the following codes: EMD-40958 (hTRPV4$_{apo}$), EMD-40959 (ARD-RhoA region), EMD-40960 (hTRPV4$_{4\alpha PDD}$ open state), EMD-40961 (hTRPV4$_{4\alpha PDD}$ closed state), EMD-40962 (hTRPV4$_{HC}$ inhibited state). The coordinates for the atomic models have been deposited in the Protein Data Bank under accession codes 8T1B (hTRPV4$_{apo}$), 8T1C (ARD-RhoA region), 8T1D (hTRPV4$_{4\alpha PDD}$ open state), 8T1E (hTRPV4$_{4\alpha PDD}$ closed state), 8T1F (hTRPV4$_{HC}$ inhibited state). hTRPV4$_{apo}$ model was built using PDB structure 7AA5 as a guide. RhoA model was built using PDB structure 1FTN as a guide. Source data are provided with this paper.

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

## Acknowledgements
We thank R. Grassucci and Z. Zhang (Columbia University Cryo-Electron Microscopy Center) for helping with microscope operation and data collection. Some of this work was performed at the Columbia University Cryo-Electron Microscopy Center. A portion of this research was supported by NIH grant U24GM129547 and performed at the PNCC at OHSU and accessed through EMSL (grid.436923.9), a DOE Office of Science User Facility sponsored by the Office of Biological and Environmental Research. A.N. is a Walter Benjamin Fellow funded by the Deutsche Forschungsgemeinschaft (DFG, German Research Foundation; 464295817). D.X.Z. is supported by the NIH (R01 HL096647). A.I.S. is supported by the NIH (R01 AR078814, R01 CA206573, R01 NS083660, R01 NS107253).

## Author contributions
K.D.N. made constructs. K.D.N. and I.A.T. prepared protein samples and carried out cryo-EM data processing. K.D.N. and A.N. prepared cryo-EM samples. K.D.N., I.A.T. and A.I.S. analyzed structural data. A.I.S. built molecular models. A.P. performed functional Ca$^{2+}$ imaging experiments. A.P. and D.X.Z. analyzed functional data. K.D.N., I.A.T. and A.I.S. wrote the manuscript, which was then edited by all authors.

## Competing interests
The authors declare no competing interests.
