## [Peer Review File · Nature Communications]

Structure of human TRPV4 in complex with GTPase RhoAREVIEWER COMMENTS

Reviewer #1 (Remarks to the Author):

The study by the Sobolevsky group examined the structural aspect of a previously discovered TRPV4-RhoA interaction, as well as the structure of TRPV4 in complex with a small molecule activator 4alpha-PDD or a small molecule inhibitor HC-067047. There are two highlights of this study. First, the TRPV4-RhoA complex formation, reported two years ago (ref#13), reveals a potentially important regulatory mechanism for a number of ankyrin repeats-containing TRP channels including but not limited to TRPV4. Specifically for TRPV4, the complex structure sheds new light on genetic mutations that causes human hereditary diseases. Second, comparison among multiple structures of TRPV4 in complex with a gating modifier offers clues for channel activation and inhibition. For these reasons, the study is of high significance. A list of issues to be addressed, mostly minor, is given below.

1. The authors claim (Introduction, lines 34-35) that the TRPV4-RhoA structure may suggest the mechanism of mechanosensitivity of the channel. TRPV4 can be activated in multiple ways addition to mechanosensitive activation. The data offered in this study do not distinguish them. Therefore, it is suggested to make a general statement about RhoA's role in regulating TRPV4.
2. Many TRP channels contain an ankyrin-repeats domain of varying length. It is well known that an ankyrin-repeats domain serves to mediate protein-protein interaction. The authors are encouraged to discuss if this and previous studies (e.g., by the Gaudet group) may suggest a general pattern of TRP channel regulation.
3. An obvious omission in this study (and a similar study deposited in BioArxiv (<https://doi.org/10.1101/2023.03.15.532784>) is comparing the TRPV4-RhoA complex structure in the presence of GTP versus GDP, which is key to start appreciating how RhoA regulates TRPV4.
4. It is mentioned that the new TRPV4 structures differ from the reported *Xenopus* TRPV4 structure; a comparison to the BioArxiv report mentioned above (also on hTRPV4 channel) indicates potential differences as well. Given that there are several TRPV4 structures of both cryo-EM and crystallography origin (e.g., doi: <https://doi.org/10.1101/2020.10.13.334797>), it would be a good time to compare their features in this manuscript.
5. Line 33: channel activation “involves” pore opening.
6. Line 146: “selectivity filter” should be “pore”.

Reviewer #2 (Remarks to the Author):

Nadezhdin et al report cryo-EM structure of human TRPV4 in complex with the GTPase RhoA, the agonist 4a-PDD, and the inhibitor HC-067047. By the comparing structures of TRPV4 in the absence and in the presence of the regulatory subunit or the ligand, the authors elucidate the modulation mechanisms of TRPV4. I would support the publication of this story after revisions.

1. Functional validation of the binding sites of RhoA, 4a-PDD, and HC-067047 by electrophysiology assays should be performed.
2. The assignment of 4a-PDD. Please improve the map quality to further confirm the density is 4a-PDD instead of lipid or detergent. In addition, please show the densities for all the ligands in the main figures.
3. Movies that show 4a-PDD and HC-067047 induced conformational change of TRPV4 would be greatly appreciated.
4. Does RhoA induce any conformational change of TRPV4?

Reviewer #3 (Remarks to the Author):

Structure of human TRPV4 in complex with GTPase RhoA

Kirill D. Nadezhdin, Irina A. Talyzina, Aravind Parthasarathy, Arthur Neuberger, David X. Zhang, Alexander I. Sobolevsky

The Transient Receptor Potential Vanilloid 4 (TRPV4) is a polymodal ion channel that responds to endogenous and synthetic small molecules, as well as downstream of cell swelling and shear stress. It regulates various physiological processes, including vascular tone, bone homeostasis, and renal function, and has been implicated in the development of neuromuscular disorders and cancer. Interestingly, the GTPase RhoA has been shown to modulate TRPV4 activity, yet structural evidence for this modulation was lacking. In this work, the authors provide structural information of the human TRPV4-RhoA complex, as well as TRPV4 structures in the presence of the agonist 4 α -PDD (open state) and the inhibitor HC-067047 (non-conductive state). The TRPV4-RhoA complex nicely highlights the interaction interface between hTRPV4 and RhoA, as well as residues at this interface that are involved in TRPV4 disease-causing mutations. Although I am enthusiastic about this work, since it provides essential and novel information to the field, this manuscript is premature in its present format. Specifically, this work does not provide any experimental data supporting the authors' conclusion regarding the mechanism of TRPV4 mechanosensation mediated by RhoA interaction with the cellular membrane, nor the mechanism by which RhoA decreases TRPV4 channel activity.

Major critiques:

1) This work does not have any experimental data supporting the authors' main conclusion. The authors claim, "Our structures suggest a mechanism of TRPV4 mechanosensation mediated by RhoA interaction with cellular membrane". From the structures and data presented here, it is unclear how the authors propose a mechanism of TRPV4 mechanosensation mediated by RhoA. The authors should consider changing this conclusion and the accompanying discussion.

2) How does the binding of RhoA decrease TRPV4 channel activity? The authors should consider increasing the manuscript's contribution to the field by providing structures of TRPV4 channelopathy mutants, such as those proposed to influence the interaction between TRPV4 and RhoA.

Minor critiques:

1) Authors should review this sentence "The narrowest point of the selectivity filter is formed by isoleucines I715" I think here they are referring to the lower gate, not the selectivity filter.

We are very thankful to Reviewers for their excellent suggestions. We have made changes in the manuscript accordingly with the details outlined in our responses below.

Reviewer #1 (Remarks to the Author):

The study by the Sobolevsky group examined the structural aspect of a previously discovered TRPV4-RhoA interaction, as well as the structure of TRPV4 in complex with a small molecule activator 4 α -PDD or a small molecule inhibitor HC-067047. There are two highlights of this study. First, the TRPV4-RhoA complex formation, reported two years ago (ref#13), reveals a potentially important regulatory mechanism for a number of ankyrin repeats-containing TRP channels including but not limited to TRPV4. Specifically for TRPV4, the complex structure sheds new light on genetic mutations that causes human hereditary diseases. Second, comparison among multiple structures of TRPV4 in complex with a gating modifier offers clues for channel activation and inhibition. For these reasons, the study is of high significance. A list of issues to be addressed, mostly minor, is given below.

We thank Reviewer #1 for the generous assessment of our work.

1. The authors claim (Introduction, lines 34-35) that the TRPV4-RhoA structure may suggest the mechanism of mechanosensitivity of the channel. TRPV4 can be activated in multiple ways addition to mechanosensitive activation. The data offered in this study do not distinguish them. Therefore, it is suggested to make a general statement about RhoA's role in regulating TRPV4.

As suggested, we have made a general statement about the role of RhoA in regulating TRPV4 and eliminated the mention of the role of RhoA in mechanosensitivity (lines 33-35).

2. Many TRP channels contain an ankyrin-repeats domain of varying length. It is well known that an ankyrin-repeats domain serves to mediate protein-protein interaction. The authors are encouraged to discuss if this and previous studies (e.g., by the Gaudet group) may suggest a general pattern of TRP channel regulation.

As suggested, we have added a discussion of TRP channel regulation through ankyrin repeat domains and referenced the previous studies by the Gaudet group (lines, 235-241).

3. An obvious omission in this study (and a similar study deposited in BioArxiv (<https://doi.org/10.1101/2023.03.15.532784>)) is comparing the TRPV4-RhoA complex structure in the presence of GTP versus GDP, which is key to start appreciating how RhoA regulates TRPV4.

We did not observe noticeable changes in our closed-state TRPV4 structures obtained in the apo conditions and in the presence of 4 α PDD and 10 mM GTPyS. Despite the excellent resolution of our maps (3.0 Å and 2.77 Å, respectively), the quality of the nucleotide density is not good enough to reliably discriminate between GDP and GTPyS. Additional experiments are required to study the effects of nucleotides on RhoA regulation of hTRPV4. However, we have now added a new Supplementary Figure 7 that demonstrates no effect of RhoA binding on TRPV4 structure. The corresponding information has been added to the text (lines 115-119, 190-194).

4. It is mentioned that the new TRPV4 structures differ from the reported *Xenopus* TRPV4 structure; a comparison to the BioArxiv report mentioned above (also on hTRPV4 channel) indicates potential

differences as well. Given that there are several TRPV4 structures of both cryo-EM and crystallography origin (e.g., doi: <https://doi.org/10.1101/2020.10.13.334797>), it would be a good time to compare their features in this manuscript.

We thank Reviewer #1 for this valuable comment. We have now updated the Supplementary Fig. 5 with additional panels that clearly illustrate differences and similarities between human and frog TRPV4 structures (doi: <https://doi.org/10.1101/2020.10.13.334797>). The corresponding information has been added to the text (lines 93-95, 804-815).

5. Line 33: channel activation “involves” pore opening.

We replaced “show that channel activation leads to pore opening” with “show that agonist binding leads to pore opening”.

6. Line 146: “selectivity filter” should be “pore”.

Thank you for noticing this obvious mistake! We corrected the text accordingly.

Reviewer #2 (Remarks to the Author):

Nadezhdin et al report cryo-EM structure of human TRPV4 in complex with the GTPase RhoA, the agonist 4a-PDD, and the inhibitor HC-067047. By the comparing structures of TRPV4 in the absence and in the presence of the regulatory subunit or the ligand, the authors elucidate the modulation mechanisms of TRPV4. I would support the publication of this story after revisions.

We thank Reviewer #2 for favorable opinion about our manuscript.

1. Functional validation of the binding sites of RhoA, 4a-PDD, and HC-067047 by electrophysiology assays should be performed.

We have made mutations at the RhoA-TRPV4 interface as well as in the binding site of 4a-PDD and HC-067047 and carried out their functional characterization, which provided strong validation of the RhoA, 4a-PDD, and HC-067047 binding sites (Supplementary Fig. 1f-h, lines 142-146, 173-178, 213-215).

2. The assignment of 4a-PDD. Please improve the map quality to further confirm the density is 4a-PDD instead of lipid or detergent. In addition, please show the densities for all the ligands in the main figures.

To improve the quality of 4a-PDD density, we performed symmetry expansion of the particles, followed by local classification and refinement. The resulting 4a-PDD density has been dramatically improved and unambiguously identifies 4a-PDD (see new panel b in Fig. 4 and panel d in Supplementary Fig. 6). We have also included densities for all the ligands in the main Figures 1, 4 and 5.

3. Movies that show 4a-PDD and HC-067047 induced conformational change of TRPV4 would be greatly appreciated.

We have made the corresponding Supplementary Videos 1-2.

4. Does RhoA induce any conformational change of TRPV4?

We have created a new Supplementary Figure 7 that demonstrates the effect of RhoA binding. To generate this figure, we conducted focused classification of particles with a mask covering one of the four TRPV4 subunits and were able to obtain two maps that represent the RhoA-bound and RhoA-unbound states. We built models for both states independently. Alignment and comparison of the RhoA-bound and RhoA-unbound models revealed neither noticeable changes in the global fold throughout the entire subunits nor local conformational changes at the TRPV4-RhoA interface. The corresponding information has been added to the text (lines 115-119, 827-829).

Reviewer #3 (Remarks to the Author):

Structure of human TRPV4 in complex with GTPase RhoA

Kirill D. Nadezhdin, Irina A. Talyzina, Aravind Parthasarathy, Arthur Neuberger, David X. Zhang, Alexander I. Sobolevsky

The Transient Receptor Potential Vanilloid 4 (TRPV4) is a polymodal ion channel that responds to endogenous and synthetic small molecules, as well as downstream of cell swelling and shear stress. It regulates various physiological processes, including vascular tone, bone homeostasis, and renal function, and has been implicated in the development of neuromuscular disorders and cancer. Interestingly, the GTPase RhoA has been shown to modulate TRPV4 activity, yet structural evidence for this modulation was lacking. In this work, the authors provide structural information of the human TRPV4-RhoA complex, as well as TRPV4 structures in the presence of the agonist 4 α -PDD (open state) and the inhibitor HC-067047 (non-conductive state). The TRPV4-RhoA complex nicely highlights the interaction interface between hTRPV4 and RhoA, as well as residues at this interface that are involved in TRPV4 disease-causing mutations. Although I am enthusiastic about this work, since it provides essential and novel information to the field, this manuscript is premature in its present format. Specifically, this work does not provide any experimental data supporting the authors' conclusion regarding the mechanism of TRPV4 mechanosensation mediated by RhoA interaction with the cellular membrane, nor the mechanism by which RhoA decreases TRPV4 channel activity.

We thank Reviewer #3 for enthusiastic opinion about our work. We agree that our work does not focus on the mechanism of TRPV4 mechanosensation but rather presents interaction of TRPV4 with RhoA as well as provides an insight into TRPV4 regulation by ligands. To avoid possible misinterpretations of our main results, we have made a general statement about the role of RhoA in regulating TRPV4 and removed the mention of the role of RhoA in mechanosensitivity from the Abstract (lines 33-35).

Major critiques:

1) This work does not have any experimental data supporting the authors' main conclusion. The authors claim, "Our structures suggest a mechanism of TRPV4 mechanosensation mediated by RhoA interaction with cellular membrane". From the structures and data presented here, it is unclear how the authors propose a mechanism of TRPV4 mechanosensation mediated by RhoA. The authors should consider changing this conclusion and the accompanying discussion.

We have edited the text of our manuscript to eliminate unsupported claims (lines 33-35).

2) How does the binding of RhoA decrease TRPV4 channel activity? The authors should consider increasing the manuscript's contribution to the field by providing structures of TRPV4 channelopathy mutants, such as those proposed to influence the interaction between TRPV4 and RhoA.

There is a paper published 2 years ago in Nature Communications that studied the effect of RhoA on TRPV4 channel activity (ref. 13), which we cite and discuss in our manuscript. We present four novel structures of wild-type human TRPV4. Each additional structure is an additional investment of significant time and resources. While we agree that it would be nice to get structures of TRPV4 with channelopathy mutations, this is a topic of a separate investigation. Besides, it is not known whether the channelopathy mutants will behave biochemically well enough to support structure determination.

Minor critiques:

1) Authors should review this sentence "The narrowest point of the selectivity filter is formed by isoleucines I715" I think here they are referring to the lower gate, not the selectivity filter.

Thank you for noticing this obvious mistake! We corrected the text accordingly.

REVIEWERS' COMMENTS

Reviewer #2 (Remarks to the Author):

The authors have addressed all my concerns and I have no further question. I support the publication of this manuscript now.

Reviewer #3 (Remarks to the Author):

Structure of human TRPV4 in complex with GTPase RhoA

Kirill D. Nadezhdin, Irina A. Talyzina, Aravind Parthasarathy, Arthur Neuberger, David X. Zhang, Alexander I. Sobolevsky

As previously mentioned, this work provides structural information on the human TRPV4-RhoA complex, as well as TRPV4 structures in the presence of the agonist 4 α -PDD (open state) and the inhibitor HC-067047 (non-conductive state). The TRPV4-RhoA complex nicely highlights the interaction interface between hTRPV4 and RhoA, as well as residues at this interface that are involved in TRPV4 disease-causing mutations.

The authors mention in the abstract that "Our structures suggest a mechanism of TRPV4 modulation by RhoA". It is unclear from the structures and data presented here which is the mechanism of TRPV4 modulation by RhoA. Supplementary Figure 7 shows that TRPV4 subunits with or without RhoA look the same. The authors should support their conclusion better or remove it from the manuscript.

I agree with the author's conclusion in lines 62-65, and they should stick to it. This work mainly reports the structure of the human TRPV4-RhoA complex and sheds light on TRPV4 activation and inhibition.

We are very thankful to Reviewers for their comments. We have changed the abstract according to the suggestion of Reviewer #3 with the details outlined below.

Reviewer #2 (Remarks to the Author):

The authors have addressed all my concerns and I have no further question. I support the publication of this manuscript now.

Reviewer #3 (Remarks to the Author):

Structure of human TRPV4 in complex with GTPase RhoA

Kirill D. Nadezhdin, Irina A. Talyzina, Aravind Parthasarathy, Arthur Neuberger, David X. Zhang, Alexander I. Sobolevsky

As previously mentioned, this work provides structural information on the human TRPV4-RhoA complex, as well as TRPV4 structures in the presence of the agonist 4 α -PDD (open state) and the inhibitor HC-067047 (non-conductive state). The TRPV4-RhoA complex nicely highlights the interaction interface between hTRPV4 and RhoA, as well as residues at this interface that are involved in TRPV4 disease-causing mutations.

The authors mention in the abstract that “Our structures suggest a mechanism of TRPV4 modulation by RhoA”. It is unclear from the structures and data presented here which is the mechanism of TRPV4 modulation by RhoA. Supplementary Figure 7 shows that TRPV4 subunits with or without RhoA look the same. The authors should support their conclusion better or remove it from the manuscript.

I agree with the author's conclusion in lines 62-65, and they should stick to it. This work mainly reports the structure of the human TRPV4-RhoA complex and sheds light on TRPV4 activation and inhibition.

The last sentence of the Abstract “Our structures suggest a mechanism of TRPV4 modulation by RhoA and provide a template for the design of future therapeutics for treatment of TRPV4-related diseases” has been replaced with “Our structures elucidate the interaction interface between hTRPV4 and RhoA, as well as residues at this interface that are involved in TRPV4 disease-causing mutations. They shed light on TRPV4 activation and inhibition and provide a template for the design of future therapeutics for treatment of TRPV4-related diseases”.